# The United Nations (UN) Card, Identity, and Negotiations of Health among Rohingya Refugees

**DOI:** 10.3390/ijerph20043385

**Published:** 2023-02-15

**Authors:** Md Mahbubur Rahman, Mohan J. Dutta

**Affiliations:** Center for Culture-Centered Approach to Research and Evaluation (CARE), School of Communication, Journalism and Marketing, Massey University, Manawatu Campus, Palmerston North 4442, New Zealand

**Keywords:** Rohingya refugee, UN card, documents, identity, health, Malaysia, New Zealand

## Abstract

Being persecuted and expelled from Myanmar, Rohingya refugees are now distributed throughout the world. The Southeast Asian nation of Malaysia has been a preferred destination for Rohingyas fleeing Myanmar’s state-sponsored genocide and more recently in a bid to change their fates from the refugee camps in Bangladesh. Refugees are one of the most vulnerable groups in Malaysia and often face dire circumstances, in which their health and wellbeing are compromised. Amidst a plethora of structural challenges, Rohingya refugees try to claim some of their rights with the aid of the UN card (UNHCR ID cards) in Malaysia. Guided by the culture-centered approach (CCA), this study examined the perspectives and experiences of healthcare among Rohingya refugees while living in Malaysia, now resettled in Aotearoa, New Zealand. The participants’ narratives showed that the UN card not only materialized their refugee status in Malaysia but also offered them a way of living in a world where documents anchor the materiality of health.

## 1. Introduction

The Rohingya, a Muslim ethnic minority group of Myanmar who fled continuous state-sponsored persecution in their motherland to seek refuge in neighboring countries, continue to face various forms of discrimination while living in neighboring countries. Starting from 1980s to date, Rohingyas living in the Rakhine state of Myanmar, as well as in Bangladeshi refugee camps, tend to travel to Malaysia to seek refuge. In addition, Rohingya women mainly tend to travel to Malaysia for marriage, as well as to join their families there [1,2]. After leaving Myanmar or the camps of Bangladesh, Rohingya refugees often seek to reach Malaysia because it has generally shown its generosity to Rohingya refugees arriving by boat or through land [3]. Moreover, most Rohingya refugees seek to travel to Malaysia because the country has similar religious beliefs and because of the relative ease of securing jobs as refugees [4]. There is also a lesser likelihood of alienation or isolation as an informally settled Rohingya community has been distributed throughout Malaysia for two to three generations [5].

Decades-long persecutions and egregious human rights violations led millions of Rohingya to flee Myanmar to neighboring countries, including Bangladesh and Malaysia, reflected in a lingering refugee situation [6,7]. According to the United Nations Refugee Agency (UNHCR), as of the end of September 2022, there were some 183,430 registered refugees and asylum seekers in Malaysia, including 157,910 refugees from Myanmar, comprising 105,870 Rohingya refugees [8]. Moreover, there are at least half a million unregistered refugees living in Malaysia [9]. Various activist groups estimate that among the unregistered refugees, several thousands may be Rohingyas. Even after arriving in Malaysia, Melanie O’Brien and Hoffstaedter argue that Rohingya refugees continue to experience cultural genocide through the processes of cultural assimilation [10]. After reaching Malaysia, the first goal of the Rohingya is to collect a UN card, as, without a UN card, they are considered illegal and do not have the opportunity to work in Malaysia.

It is critical to note here that the stateless Rohingya coming from Myanmar or Bangladesh do not have any identity documents. Against this backdrop, the UN card serves as a lifeline for refugees living in Malaysia. In this manuscript, we report from our academic–activist–advocacy work with Rohingya refugees living in Aotearoa, New Zealand, who came from Malaysia under a third country resettlement process. Drawing upon the framework of the culture-centered approach (CCA) and based on in-depth interviews with 26 Rohingya refugees, we depict the role of the UN card as the basis for securing health among refugees. The Rohingya participants express their everyday lived experiences in Malaysia, articulating the role of the UN card as central to the everyday meaning of health when living in Malaysia. Rohingya refugee participants also articulate that those Rohingya who have been living in Malaysia for several years still rely on the UN card as their only identity document to somehow protect their rights and fulfill the basic requirements of life. Rohingya lost their citizenship rights in Myanmar in 1982 through the enactment of apartheid, such as the Citizenship Law, introduced by the then military government. Although the Rohingya community has been living in Myanmar since the 8th century, the denial of their identity as citizens forms the basis of the violence directed at the Rohingya people [11,12,13,14]. As a result of the Citizenship Law, after 1982, the majority of Rohingya did not have any identity documents and became stateless. For most of the Rohingya (born after 1982) living in Malaysia, the UN card was the first ever document that they received in their lives [15]. Nursyazwani discusses the concept of “mobile refugees”, suggesting that Rohingya refugees in Malaysia practice “imaginary citizenship”, drawing on the UN refugee card to claim their rights, creating pathways to recognition as political subjects. As emergent from the in-depth interviews that we carried out, the UN card serves as a basis in claiming the right to health. In this manuscript, we examine the role of access to documents (papers) as the basis for securing health, asking the following: how do Rohingya refugees construct the role of documents (papers) in their negotiations of everyday health? We depict the ways in which the refugee participants articulate the UN card as a basis for securing health and wellbeing, pointing to the importance of health communication as advocacy that works to create document/paper-based registers for identity through which refugees can be recognized as legitimate participants in securing the right to health.

## 2. Materials and Methods

### 2.1. Data Collection and Analysis

This article reports from 26 in-depth interviews of Rohingya refugees once living in Malaysia but now resettled in Aotearoa, New Zealand. The sample included 20 Rohingya male and 6 Rohingya female participants, aged between 18 and 60 years. The participants were recruited using the purposive sampling method, attending to the guiding questions, “Which voices are not represented here?” and “How do we invite these voices in?”, while continually attending to the intersectional erasures in the discursive space. We drew upon the snowball sampling strategy, dialoguing with Rohingya community researchers and the participants in reaching out to additional participants, guided by the two questions outlined above, which sought to engage with those at the margins of the community. All of them had lived in Malaysia for at least five (5) years and then secured the opportunity for third country resettlement through UNHCR, arriving in New Zealand between 2017 and 2021. The interview protocols were designed using the conceptual framework of the culture-centered approach (CCA), based on a guiding framework that explored the interplays of health with culture, structure, and agency, and co-created with Rohingya community researchers [15]. The study was peer-reviewed and considered of low risk under the human ethics guidelines of Massey University. Participants were provided with information sheets and consent forms to sign, and they were given the option of reviewing their anonymized interview scripts (the researcher read the documents in front of them for those who were unable to read).

Between September and November 2021, we carried out the 26 individual in-depth interviews in the Palmerston North area of New Zealand, each interview lasting between 30 min and 90 min. The in-depth interviews were conducted by both the authors, with the help of a community researcher, who was also a member of the Rohingya refugee community of New Zealand. All the interviews were conducted in the Rohingya language and were tape-recorded with the participants’ consent. Then, the interviews were translated and transcribed by the first author, and the transcripts were double-checked by the second author. We drew upon co-constructivist grounded theory to analyze the interviews, placing the emergent codes in dialogue with the concepts of culture, structure, and agency outlined in the CCA [15]. The interviews were first coded line-by-line, following the open coding process to identify concepts/themes before forming relationships between the concepts and creating categories (axial codes), and then providing theoretical integration to form core categories (selective codes). For instance, the open codes “paperwork for the UN card”, “NGO referrals”, “going into jails/detention centers to get the UN card” were coded under “process of getting the UN card in Malaysia”. The various elements related to UN card and health, i.e., (1) securing livelihood with the UN card; (2) lower treatment costs with the UN card, (3) some levels of protection with the UN card, (4) UN card for child education and (5) UN card for third country resettlement, were categorized under the core category of “UN card”. The initial coding led to 231 open codes, which resulted in 47 axial codes and nine selective codes. This manuscript reports on the selective code of “UN card”, which comprises the five axial codes identified above.

### 2.2. Data Collection Method and the CCA

The CCA is a meta-theoretical framework for conceptualizing health and communication at the margins, situating health at the intersections of culture, structure, and agency, with the voices of participants at the margins (such as Rohingya refugees), whose voices have historically been erased by dominant discourses offering the conceptual nodes [15]. In the CCA, cultural contexts offer entry points for the generation of theoretical insights that describe how health decisions and meanings are negotiated in communities at the margins [15]. Agency is expressed through cultural meanings, situated in a relationship with structures that constrain or liberate the process of fulfilling the health needs of any community [16]. In addition to its three primary constructs (culture, structure, and agency), the CCA includes several secondary features—including, for example, voice and dialogue, context and space, values and criticism, position and place—that outline the process of the dialogic co-construction of health with the “margins of the margins” community [17].

The primary goal of the CCA is to co-create spaces to listen to traditionally silenced voices (in this instance, Rohingya refugee voices) in constructing meaningful and culturally relevant health solutions [18]. Dialogue is an important element of the CCA, through which the members of the marginalized community express their thoughts so that the local meanings of health can be articulated and understood, continually negotiating the spaces between the local contexts and global frameworks for the organization of health [19,20,21,22]. Listening to the voices of marginalized communities is seen as an opening to build opportunities for structural transformation by grounding the development of health solutions in local interpretations. The conceptual frameworks and the methodological tools emergent in the CCA serve toward co-creating community-driven solutions [20,21,22,23,24,25]. The CCA opens the definition, meaning, and design of participation to community voices (Rohingya refugees), with the goal of constructing health-related “theories from below” [16,26,27]. The dialogue with the participants begins with the question, “What does health mean to you?”, followed by questions such as “What are the challenges to health you experience?” and “How do you negotiate these challenges?” We note here that the open-ended conversation on everyday meanings of, challenges to and solutions to health led the initial dialogues toward the role of papers, where the Rohingya participants noted the role of papers, and particularly the UN card, in securing access to health and mobility. Subsequently, we modified the interview protocol, based on dialogues with the Rohingya community researcher, to include questions related to the UN card when in Malaysia.

### 2.3. Process of Obtaining UN Card in Malaysia

The formal UN cards started to be distributed to Rohingya refugees in Malaysia in the early 2000s. There is no standard process to obtain the UN card from the UNHCR, Malaysia. Generally, refugees are encouraged to fax or post a letter with their biodata (full name, date and place of birth, ethnicity) and a photograph to the UNHCR to be registered first and then obtain the UN card. As a large number of Rohingya are illiterate and cannot send their biodata for registration, they directly travel to the UN Office for registration. A refugee can be registered through referrals from NGOs as asylum seekers. Moreover, during UNHCR visits to immigration detention centers, refugees can be registered. The UNHCR started the Rohingya refugee registration process in Malaysia in the 1990s, but, at that time, formal UN cards were not given to the refugees. A Rohingya refugee living in Malaysia for 27 years observed,

“Then in 1992, the UN opened the registration process for the Rohingyas. At that time, they did not offer us a UN card but just a paper containing your name, father’s name, and date of birth. After one month you need to go again for renewal. At that time, it was easy to register. In 1992 there were only about 400 Rohingyas living in Malaysia. At that time, you just go to the UN Office, and you can register yourself. But nowadays it is very difficult to get a UN card.”(Rohingya male who stayed Malaysia for 27 years)

Another Rohingya refugee who lived in Malaysia for 22 years mentioned,

“In 1995 I entered Malaysia through Bangladesh, then India, and then Thailand, and lastly Malaysia. But I got the UN card in 2003 because after my arrival in Malaysia in 1995, the UNHCR did not offer us the UN cards. So, from 1995–2003, I spent my life in Malaysia as an unregistered refugee.”(Rohingya male who stayed in Malaysia for 22 years)

Rohingya refugee participants mentioned that, before 2010, it was easy to obtain UN cards, but, currently, they face too many difficulties to obtain a UN card. In recent years, to acquire a UN card easily, Rohingya participants noted following the process of entering jail/immigration detention centers. Without being arrested by the immigration police, a refugee cannot enter a detention center, and so some Rohingya are intentionally arrested by the Malay police so that they may be able to obtain a UN card. A Rohingya refugee who stayed in Malaysia for 5 years shared his experience of obtaining a UN card:

“Before 2010, it was easy to get the UN card but after that it was very tough to get the UN card in Malaysia. Before 2010, you may directly go to the UN office and apply for the UN card. But now it is very hard to get the UN card. But the police custody system helps a Rohingya to get the UN card very easily. So, like me, some Rohingyas wish to be arrested by the Malay Police so that they will be able to get UN cards.”(Rohingya male who stayed in Malaysia for 5 years)

A Rohingya refugee who received his UN card after being caught by Malay police shared,

“One month after my arrival in Malaysia, me and my brother were caught by the police and the police sent us to jail. We spent 7 months in jail. Not only me and my brother, but there were also 12 other Rohingyas caught by the police at that time. After 7 months in jail, we were sent to the immigration camp. At that time UN people came to help us. The UN people then interviewed us. When they confirmed that we are Rohingya then they processed our UN cards.”(Rohingya male who stayed Malaysia for 7 years)

After the birth of a Rohingya child in Malaysia, the child is registered by the UNHCR. To obtain a UN card for the new-born, the child needs to be taken to the UN Office; the UN staff confirm the UN card of his/her father and allow the registration of the new-born. The Rohingya women who do not have UN cards in Malaysia generally obtain UN cards when the new-born child receives their registration at the UNHCR. Two Rohingya women who lived in Malaysia for 9 years and 7 years, respectively, shared their experiences:

“In 2014–15, I got the UN card. To get the UN card, my husband and I had to go 6 to 7 times to the UN Office. After the birth of the first child in Malaysia, we went to the UN to get the UN card. They took my baby’s picture to give him the card and at that time my husband mentioned that the baby’s mother still does not get the card and then I got the UN card. Generally, the Rohingya women get the UN card in Malaysia after the birth of a Rohingya child.”(Rohingya female living in Malaysia for 9 years)

“Though I have two children in Myanmar, but after the birth of a child in Malaysia, I got the UN card. I along with my child went to the UN office to get the UN card. The UN people took my baby’s picture to give him the card and at that time, I also got the UN card.”(Rohingya female living in Malaysia for 7 years)

## 3. Results

As per the articulations offered by the Rohingya participants, five overarching themes were rooted in the everyday negotiations of health and simultaneously depicted the role of the UN card in their negotiations of livelihood in Malaysia: (1) securing livelihood with the UN card, (2) lower treatment costs with the UN card, (3) some level of protection with the UN card, (4) UN card for child education and (5) UN card for third country resettlement. Note here that although themes 1, 3, 4 and 5 are not directly reflected as health, in the voices of the refugee participants, their access to a livelihood, protections against police harassment, the education of children and opportunities for resettlement are intertwined with the everyday meanings of health.

### 3.1. Securing a Livelihood with the UN Card

For all the participants, everyday understandings of health were rooted in the capacity to secure a livelihood. As Malaysia is not a party to the 1951 refugee convention or its 1967 protocol, refugees who end up in Malaysia are considered as “undocumented migrants” and afforded no legal rights. However, as with other refugees in Malaysia, Rohingya refugees can perform various jobs to manage their livelihoods if they have UN cards. Rohingya refugees are engaged in the agricultural and construction sectors in Malaysia, where there is a higher demand for low-skilled workers. These jobs include extreme manual labor with limited labor rights and with little to no pay, which in turn leads to mental and physical trauma experienced by the refugees [28]. A Rohingya refugee who lived in Malaysia for 10 years observed,

“In Malaysia not only Rohingyas but also many other refugees are living… They have their passports, but we do not have any documents. The only document we can show the employer is the UN card. Many refugees come to Malaysia as they could easily find a job. The employer even does not ask anything except the UN card and sometimes not even the UN card. Rather they are happy as they can recruit the refugees at a lower cost. If our salary is 1000 ringgit, then the employer needs to pay Malay people 3 times, i.e., 3000 Ringgit. As a result, the refugee people do easily get jobs in Malaysia.”(Rohingya male who lived in Malaysia for 10 years)

The participants articulated that after obtaining a UN card, they found it easier to secure work in Malaysia. One of the participants noted, “UN card does not give us the permission to do the job but still, we find more jobs in Malaysia and get the salary at the end of each day.” Participants shared the role of access to a UN card in securing work.

“I did various jobs in Malaysia. There is no scarcity of jobs in Malaysia. Construction work, labour work or even buying and selling fishes etc. were done by me. I did not even stay idle for a day without a job. Various kinds of jobs are available in Malaysia till today. No person is jobless in Malaysia, and we get the job payment every day in the evening. But without a UN card, you will not get the job.”(Rohingya male who lived in Malaysia for 17 years)

Although the UN card is seen as the necessary paperwork that supports the participants in securing work, a number of participants pointed to the limited role of the card in offering protections and access to a secure livelihood.

“We were treated as foreigners in Malaysia. Though we had UN cards, we did not get official permission to work there. We had to work in Malaysia but that was not officially permitted and so we were always scared. Malaysian police always disturbed us. The Police mentioned that the UN card does not permit you to do a job. They told us that we could not do our jobs. Then we had to manage by giving a bribe to the police.”(Rohingya male who lived in Malaysia for 14 years)

Note here the ambiguity around the role of the UN card. Although the participants pointed to the UN card as an enabler in securing access to work, they simultaneously noted that the UN card does not provide official permission to work. This ambiguity around the UN card shaped participants’ harassment in the hands of the police. The sense of being scared and feeling anxious was communicated across the interviews, and this was intertwined with everyday meanings of health. Moreover, everyday acts such as offering bribes to the police emerged as strategies for survival amidst the ambiguity and the harassment.

### 3.2. Lower Treatment Costs with the UN Card

In Malaysia, non-citizens are given access to all government hospitals and clinics, but they are charged a non-subsidized rate [26,27,28,29,30]. Meanwhile, all registered (UN card holder) refugees and asylum seekers (having asylum-seeker letters) can access treatment facilities at all public health facilities and enjoy a 50% subsidized rate of treatment [28,29,30]. Moreover, refugees in Malaysia (registered or unregistered) experience substantial barriers in accessing healthcare because of language barriers, cultural differences and a sense of anxiety about accessing the healthcare system. Amidst the difficult and life-risking journeys during their travel, mainly by boat, and the poor health conditions exacerbated amidst the politics of violence in Myanmar, Rohingya refugees often experience acute and immediate health crises upon arriving at their country of destination, including in Malaysia. Upon arrival in Malaysia, many of them cannot access any health facilities, being treated as unregistered refugees and not having any money with them. One of the Rohingya refugee participants shared her experience:
“No complaint from a refugee (registered or unregistered) is generally allowed in Malaysia. Another thing when we visit any hospital, our fee is 100 times more than the fee of a Malay resident. Again, after taking 100 times the fee, sometimes they give only 10–20 Paracetamol. This is for the Rohingya refugees who have UN cards, and for those who do not have UN cards, they have to pay 200 times.”(Rohingya female who stayed in Malaysia for 06 years)
One of the participants who had lived in Malaysia for 14 years mentioned, “When we were sick, we had to pay more money in the hospitals. Even the medical personnel asked us- ‘why do you come to our country?’” While describing his experience in Malaysia with racial discrimination in healthcare, one of the participants mentioned,

“When my wife was pregnant, she went to the clinic for a checkup. But at that time doctors did not behave well with her. They asked my wife, “why do you give birth to children as you do not have any legal documents?” Though we are paying more than double still they show their hatred towards us. When they do any diagnostic test like blood check-up, urine checkups they show their hatred towards us by their attitudes, and they take more than double fees from us.”

Many participants pointed out financial difficulties as a key barrier in accessing healthcare services in Malaysia. As Rohingya refugees are allowed only to perform low-grade jobs because of availability, they are unable to earn more in Malaysia. A participant mentioned, “in a day, the Rohingya refugee can earn 50–100 Malaysian Ringgit. Then how can we avail better healthcare services in Malaysia?” Most participants proposed that if the Malaysian government had granted refugees and asylum seekers the basic right to employment by integrating them into the formal workforce, it may have improved the refugees’ earning power, thus enabling them to afford better healthcare services. This financial precariousness is considered amidst the struggles with health that are shaped by the boat journey.

“Due to the problems faced by me during the boat journey, I became sick after reaching Malaysia. During about three months boat journey, I could not eat properly, could not use the toilet properly, and had to sleep in a crowded space in the boat. We could not drink fresh water. Instead, we had to drink the salty water of the sea. As a result, I became sick and for the first nine months stay in Kuala Lumpur (Malaysia), I could not do any job and at that time, I was unregistered as well.”(Rohingya male who stayed in Malaysia for 05 years)

Unregistered Rohingya can only visit NGO clinics and some private clinics in Malaysia. When they are unregistered or even registered, the cost of healthcare is unaffordable for many of them [26]. Unregistered or even registered Rohingya refugees are afraid of being arrested by the Malaysian police and so do not travel to seek medical services. One of the participants mentioned that when he went to do some shopping for groceries shortly before arriving in New Zealand, the Malay police harassed him and retained the travel bag that he had purchased for travelling.

Against this backdrop of fear of police harassment, Rohingya refugees in Malaysia are eager to obtain UN cards to help them to secure a legal basis to access healthcare facilities with 50% subsidized medical costs in public healthcare settings.

“Generally, without the UN card we cannot visit the doctors in Malaysia. So, during my pregnancy period in Malaysia I faced a lot of problems visiting a doctor. You may be able to visit the doctor without the UN card then you need to pay more than the double amount. With the UN card still we had to pay more than the Malay people. But the privilege was with the UN card if the payment of the hospital is 200 Malaysian Ringgit, then you have to pay 50%, i.e., 100 and the rest 50% is paid by the UN/Govt.”(Rohingya female who stayed in Malaysia for 6 years)

“If a Malay resident gives 1 Malay Ringgit, then we have to give 100 Malay Ringgit. It is just the doctors’ fee without any medicine cost. But in the private clinics in Malaysia, all (foreigners and Malay people) have to give the same fee but more treatment cost. In government hospital in Malaysia for Malay people if the treatment cost is 1 dollar Malaysian money then for a Rohingya refugee it is 100 dollar Malaysian money. But there is one thing, if you have a UN card in Malaysia then 50% of your treatment cost is given by the UN and 50% needs to be borne by you.”(Rohingya male who stayed in Malaysia for 11 years)

“My child was born in Malaysia. But the childbirth fee was too much in Malaysia. We had to pay 7000 (seven thousand) Malaysian Ringgit during the birth of my child as at that time I did not have the UN card. As I did not have the UN card, we also faced problems finding hospitals. The Malay police also harassed my husband several times.”(Rohingya female who stayed in Malaysia for 10 years)

### 3.3. Some Level of Protection with UN Card

Both unregistered and registered Rohingya refugees have been living in Malaysia with a constant fear of arrest, detention, and even deportation, and these negative registers of affect shape their everyday meanings of health and wellbeing. Rohingyas who are registered with the UNHCR (UN card holders) obtain some level of informal protection against arrest, detention, and deportation; unregistered Rohingya are mostly excluded from any protection offered by the UNHCR [25]. The Rohingya participants who entered Malaysia in the 1990s stated that they had been deported to Thailand or Myanmar on a couple of occasions and they had been detained several times by the police or immigration authorities of Malaysia.

“When I entered Malaysia in 1989 at that time, no UN party was there. Sometimes, we were caught by the immigration police, and they pushed us back to Thailand. Then after giving money to the Dalal (broker), we entered Malaysia. But the problem was that all the pushed back persons could not come back to Malaysia. Some may have died on the way. I was also caught by the Malay police 4–5 times and pushed me back to Thailand. It was life at that time.”(Rohingya male who lived in Malaysia for 29 years)

“Sometimes, they (Malaysian authorities) sent us near the Myanmar border. Again, we struggled to come back to Malaysia as we did not have any documents where we could go. In Thailand there was less opportunity for work and so we all tried to come back to Malaysia to get a job.”(Rohingya male who lived in Malaysia for 29 years)

The Rohingya participants mentioned that after obtaining UN cards, although they had been harassed by the Malaysian police, they received some protection. For example, even when the Malay police arrested them, with the help of their UN cards, the Rohingya refugees could secure their release from jail or detention centers within a short period of time.

“After getting the UN card, I was also arrested by the Malay Police. Then they again put me in the immigration camp. But this time, I could be released within a short period of time as I have had the UN card. When the police got confirmation of my UN card, they released me.”(Rohingya male who lived in Malaysia for 11 years)

“So many times, the Malay police tried to harass me. As I was driving my motorcycle, the police always tried to disturb me. As a refugee, we were not allowed to take driving licenses in Malaysia and for this when any police saw me, then he wanted some bribe. As a refugee, we were not also allowed to open any Bank account in Malaysia or even not allowed to rent a house. As I was driving the motorbike so many times, I had to give bribes to the police. But if you do not have the UN card then you could not manage the police and would have to go to jail.”(Rohingya male who lived in Malaysia for 10 years)

### 3.4. UN Card for Rohingya Child Education

Rohingya children are not allowed to study in state schools in Malaysia. Whilst most have no access to education, a few Rohingya children can study in UNHCR-run and NGO- or community-based schools. However, to attend these schools, a UN card is mandatory. Most participants spoke about the importance of granting their children access to formal education in Malaysia. The anxiety around not being able to send their children to school shaped everyday meanings of health. They also mentioned that they applied for third country resettlement from Malaysia, mainly concerned with the futures of their children.

“As refugees we always faced problems in Malaysia. We cannot move freely, we cannot rent any house, our children are not allowed to get education. We always think that without education the kids do not have any future. We understand the importance of education as in our life we did have any opportunity of education. Child education was one of the reasons to apply for third country settlement.”(Rohingya female who stayed in Malaysia for 6 years)

“The UN card does not give us any rights to Malaysia. We can only do our jobs as laborers or any low category jobs, but the UN card does not give us or our children citizenship rights in Malaysia. Even our children do not get Govt. education facilities by the UN card. But with the UN card the children can study only in the UN run school but that is not the formal education.”(Rohingya male who stayed in Malaysia for 10 years)

“When the children are going to school in Malaysia, we face problems. When the children are in class I, no problem but when the kids are in class III, the Malaysian Govt said, the children of refugees could not study further. With the UN card the children can study up to class III or IV. Then we were in tension. Yes, we had many jobs in Malaysia, but the children do not have any future as they could not study more.”(Rohingya male who stayed in Malaysia for 25 years)

### 3.5. UN Card for Third Country Resettlement

Resettlement is the transfer of refugees through the UNHCR from an asylum-offering country (Malaysia) to another developed state (Europe, USA, Canada, Australia, New Zealand, etc.) that has agreed to admit them and ultimately grant them permanent residence. Resettlement is the preferred solution for refugees living in Malaysia, as the two other refugee solutions, repatriation and local integration, are impossible in this country [29,30,31]. Resettlement to a third country is the ideal option for many Rohingya refugees, and many of them travel to Malaysia with the intention of obtaining third country resettlement. Rohingya refugees are generally not interested in travelling to their homeland of Myanmar as the situation is not still resolved and they fear being persecuted again. Securing access to a UN card in Malaysia enables them to achieve third country resettlement. However, without having been registered with the UNHCR (UN card holders), Rohingya refugees are not entitled to apply for the third country resettlement process. During the interviews, the Rohingya refugees presented their narratives in initiating the third country resettlement process in Malaysia as a basis for health.

“From 2011, I started to apply to the UN for my third country settlement process. I requested them to help us to send me to any third country. I was called in 2012 for an interview. Then my family member has been called with me by the UN as all of my family members had the UN cards. Then they interviewed me several times. Then the medical test of my family members has been done. Then at last in 2018, I got the opportunity to come to New Zealand. So, about 6 years have been required to finalize my third country settlement process.”(Rohingya male who lived in Malaysia for 29 years)

“After some months of my application, the UN called me and took my interview. They asked me what problems I faced in Malaysia. I answered all the questions they asked me. I mentioned I could not lead a good life here in Malaysia, faced the problem of giving the house rent, faced the problem of giving the doctors fee etc. I again in person request them to send me abroad. But without the UN card we cannot apply for the third country settlement.”(Rohingya male who lived in Malaysia for 15 years)

“No such problem to apply for the third country settlement as we had UN cards. The main problem was that we did not have any car and sometimes we faced difficulty getting the bus to go to the UN Office. After giving the interviews we had to complete the medical test. But then we had to wait for 3 years after applying to the UN to get the final resettlement in New Zealand.”(Rohingya female who lived in Malaysia for 6 years)

## 4. Discussion

This qualitative study examines the meanings of health and key challenges in addressing health among Rohingya refugees while they have been living in Malaysia. Stateless people, such as the Rohingya refugees in Malaysia, without having any state-based identity documents such as passports, driving licenses, or birth certificates, are marked as suspicious bodies who are illegal in the country. This marker of (il)legality shapes their everyday experiences of health and wellbeing. When asked, “What does health mean to you?”, the participants highlighted an array of lived experiences and struggles that are tied to the struggles of securing an identity document. Their everyday challenges with regard to health are intertwined with anxieties and insecurities around securing a UN card.

The participants’ narratives put forth the concept of “negotiated mobility”, suggesting that refugee health is constituted amidst the negotiations of mobility that are tied to documents. Participants’ voices rendered legible the organizing role of documents as a basis for organizing health. Against this backdrop, the negotiations of health by Rohingya refugees turn to the processes around securing a UN card. Negotiating access to documents is the basis for securing access to health and wellbeing. Rohingya refugees can register themselves with the UNHCR to obtain UN cards, and this securing of the document is articulated as an anchor in their meanings of health. Although the UN card is limited in scope and the participants note that they do not receive much assistance with the card, voicing that they cannot work legally and face limited access to education, healthcare, and other social services in Malaysia, the card nevertheless serves as the basis for claiming an identity, which is at the heart of securing health [27,28,29,30,31,32]. Rohingya refugees want to obtain a UN card as soon as possible when they reach Malaysia, as, without a UN card, they become illegal in the country. This negotiated mobility shifts back and forth between the impossibility and possibility of mobility among Rohingya refugees in Malaysia. On one hand, the impossibility of mobility is shaped by the lack of access to documents, the violent erasure of which has shaped the violence experienced by Rohingya refugees. On the other hand, securing access to documents shapes the ongoing negotiation of mobility, including access to health, education, and pathways to residency in a third country, which are seen by the participants as integral to their health and wellbeing.

Rohingya refugees living in Malaysia view UN cards as their most important document of protection and the only form of defense that they have against detention, deportation, and police violence. It also helps them to find jobs, secure affordable access to healthcare facilities in a subsidized manner, and educate their children in informal educational institutes. All the Rohingya participants considered the UN card as the main document by which they could access public health facilities and other services in Malaysia, as, without a UN card, medical personnel do not treat them well.

All the participants recognized that their access to facilities in Malaysia was possible due to the UN cards that they had with them. Without a UN card in Malaysia, the Rohingya are unable to stay, to obtain jobs, to avoid harassment in the hands of the police, to access public healthcare facilities and to obtain access to education for their children. They also mentioned that without UN cards, they would not have been able to migrate to New Zealand through the third country resettlement process. When asked about the meaning of health, many participants replied, “When I can live well that is health to me”, and living well is rooted in securing a UN card. Most Rohingya participants confirmed that without a UN card (or being unregistered), it is not possible to live well in Malaysia, and so the UN card is the meaning of health among them. The study highlights the role of documents as a basis for securing health, raising critical questions about the ways in which documents are tied to borders. In the context of refugee health, challenges of violence, trauma, and an inability to access basic health services point to the need for theorizing and collective action that critically interrogate and attend to the organization of borders. Borders are organized and mobilized to expel refugees and carry out violence, and simultaneously to keep them out and deny them access to fundamental rights such as healthcare by marking them as illegal. Building infrastructures for refugee health at the margins fundamentally ought to be organized around dismantling the overarching ideology of borders and citizenship as the organizing features of healthcare.

This study has some limitations. First, we interviewed refugees who had moved through Malaysia into New Zealand through the third country resettlement program. The strength of the analysis, therefore, is the depth of insight secured by participants who had already moved through the third country resettlement process. However, one of the limitations of the current study is that we did not carry out in-depth interviews with refugees in Malaysia. Future scholarship ought to consider a comparative framework, examining Rohingya refugee experiences across contexts. Moreover, triangulating the data with the experiences of Rohingya refugees in Malaysia would further strengthen the findings.

The findings draw attention to the role of the identity document as an anchor to the meaning of health among refugees at the “margins of the margins”. For undocumented refugees who have been fundamentally erased through the removal of access to citizenship and accompanying forms of identification documents, the UN card offers access and mobility. Moreover, the card emerges as the basis for securing fundamental rights, including protection from harassment by police. Health communication interventions addressing the health of refugees have key roles to play in advocating for UN cards as identity documents, and in securing the rights of refugees to UN cards.

## 5. Conclusions

Traumatic pre-flight (persecutions in Myanmar), flight (journey by boat risking lives) and post-flight (living in unhealthy conditions) circumstances at the country of asylum (Malaysia) constitute the meanings of health among Rohingya refugees living in Malaysia and migrating subsequently to other parts of the globe. Our study identified a wide range of factors that constitute meanings of health for Rohingya in Malaysia, vitally anchored in securing access to a UN card. Our study highlights that the key barriers to healthcare access for Rohingya refugees in Malaysia include unhygienic living conditions, legal and protection issues, and the inability to afford healthcare. Recognizing these barriers, still, Rohingyas are interested in living in Malaysia, as the country provides them with better protection compared to other asylum-providing countries such as Bangladesh, India, Pakistan, Indonesia, or Thailand. Nowadays, the most desirable opportunity for Rohingya refugees moving through Malaysia is the third country resettlement option, by which they can live peacefully in a developed country. Moreover, access to fundamental resources such as education and subsidized healthcare among Rohingya refugees is tied to the UN card. Therefore, the UN card emerges as the anchor to meanings of health or meanings of a good life for Rohingya refugees living in Malaysia, shaping refugee health communication as health activism to secure documents as the basis for accessing health.

## Data Availability

Data is not available to protect the identity of the participants.

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
