# Peer review of "The United Nations (UN) Card, Identity, and Negotiations of Health among Rohingya Refugees"

_ijerph, 2023, doi:10.3390/ijerph20043385_

Round 1

Reviewer 1 Report

The paper is very, very descriptive, it lacks in a clear research questions and it is not embedded in any theoretical debate. What would really render itself as a useful anchoring point to align the collected empirical data to, is to discuss the materiality and symbolic significance of papers, including refugee cards.

This has been done before, very beautifully indeed, by : Nursyazwani in her essay on: "Mobile Refugee: Rohingya Refugees’ Practices of Imaginary Citizenship in Klang Valley, Malaysia".

This is cited in the references but there is zero engagement with the really interesting argument out out by Nursyazwan.

In rergard to Rohingya there is a rich body of literature, yet many core readings have been dismissed by the authors here (Hoffstaedter, Huennekes) and  I wonder why.

In any case, i think the paper needs more theoretical grounding to be relevant for a broader readership. I recommend: Paper Trails (ed. volume)

Author Response

Thank you so much for your detailed and constructive comments regarding the manuscript. We appreciate your guidance. Below we outline the revisions we made to the manuscript, which we have highlighted.

A) The paper is very, very descriptive, it lacks in a clear research questions and it is not embedded in any theoretical debate. What would really render itself as a useful anchoring point to align the collected empirical data to, is to discuss the materiality and symbolic significance of papers, including refugee cards.

This has been done before, very beautifully indeed, by : Nursyazwani in her essay on: "Mobile Refugee: Rohingya Refugees’ Practices of Imaginary Citizenship in Klang Valley, Malaysia".

This is cited in the references but there is zero engagement with the really interesting argument out out by Nursyazwan.

Thank you for encouraging us to clearly articulate a research question and anchor the empirical observations in theory. Thanks also for encouraging us to engage with the important work of Nursyazawan. We have now done so and delineated our theoretical contribution, explaining in depth the theoretical contribution made by this manuscript. We have offered the conceptual framework of negotiated mobility as the basis for health, adding to the current theorizing on Rohingya health.

B) In rergard to Rohingya there is a rich body of literature, yet many core readings have been dismissed by the authors here (Hoffstaedter, Huennekes) and  I wonder why.

We have now drawn in additional literature, including the literature suggested. Earlier, we were selective about the literature included because of the length. In the revised version, we have drawn in more literature and connected it to the manuscript.

C] In any case, i think the paper needs more theoretical grounding to be relevant for a broader readership. I recommend: Paper Trails (ed. volume)

We have drawn in the literature to outline and theorize the role of access to documents as the basis of access to health.

Reviewer 2 Report

Overall paper comment: The CCA method helps broaden definitions of "health" by using ground-up, participant-centric interviews and data. This paper is well structured, with all 5 themes having a clear section with qualitative data quotes. This global work aims to show how negative registers of affect and of identity shape health perceptions in Rohingyas in New Zealand while they were refugees in Malaysia. While I have no doubt about the thesis or its validity, there are a few omissions that, if corrected, might help improve this manuscript.

Methods:

1.       How were the 26 recruited, and what does this mean for generalizability to all Rohingyas in Malaysia?

2.       Lines 93-95: I am not familiar with the “microscope method” in particular. Provide a brief description for readers.

3.       What other methods of data review and coding were used?

Results:

1.       OK, five themes are noted. I am still unclear how these were selected from the undoubtedly many themes that arose. It would be helpful to either list all themes, or if this is too onerous in table form, to mention this in the results to let the reader know how many of the 26 interviewed mentioned each theme and why the authors selected (or how the data selected) the top 5. Without this, we have interesting 5 themes chosen and described enough, but with no way to assess face validity of this study.

2.       Para starting with line 244: This P has almost nothing to do with the UN card. Either remove or describe a bit why it is relevant.

3.       End of section 3.3: must mention that, on the other hand, being detained is HOW they have obtained the UN card.

Discussion and/or other key points:

1.       Is there any validation of the story that it is more costly for healthcare (or true for any of the other 5 themes for that matter) for non-UN card holders? To be scientifically honest, it will help readers to understand the correlation between perception and accuracy. As another example, the validity of the belief that the UN card will get them to resettlement faster, than say applying for asylum immediately on arrival to Malaysia, is not explored. It sounds like, even with the UN card the process is long. I think they want it to be shorter, which has nothing to do with getting the UN card.

2.       Limitations: lack of showing the structure of all themes and a selection bias is highly likely. Also, these authors have published on their method and on this topic. How does this paper add to the good work the authors have already published?

Other specific comments:

1.       Lines 70-71 and 73-74 Redundant message, though clear

2.       Starting line 398: The next three paragraphs are data and are redundant to the themes. If valuable, they should be in “Results.”

Author Response

Thank you so much for your encouraging, thoughtful, and constructive feedback on the manuscript. We greatly appreciate the detailed insights and helpful suggestions you have offered. Below we present the changes we have made to the manuscript, that are highlighted in the manuscript.

Overall paper comment: The CCA method helps broaden definitions of "health" by using ground-up, participant-centric interviews and data. This paper is well structured, with all 5 themes having a clear section with qualitative data quotes. This global work aims to show how negative registers of affect and of identity shape health perceptions in Rohingyas in New Zealand while they were refugees in Malaysia. While I have no doubt about the thesis or its validity, there are a few omissions that, if corrected, might help improve this manuscript.

Thank you once again for seeing the value in this manuscript and for your thoughtful feedback. Thanks for identifying the key strengths of the manuscript, which helped us crystallize our contribution and rework the discussion.

Methods:

  1. How were the 26 recruited, and what does this mean for generalizability to all Rohingyas in Malaysia?

Thank you. We have described this in the methods section.

  1. Lines 93-95: I am not familiar with the “microscope method” in particular. Provide a brief description for readers.

Thank you. We have now provided more depth about the analysis process, walking the reader through the steps.

  1. What other methods of data review and coding were used?

We have described the grounded theory framework that we used and the steps of the process. We have also descrived validity.

Results:

  1. OK, five themes are noted. I am still unclear how these were selected from the undoubtedly many themes that arose. It would be helpful to either list all themes, or if this is too onerous in table form, to mention this in the results to let the reader know how many of the 26 interviewed mentioned each theme and why the authors selected (or how the data selected) the top 5. Without this, we have interesting 5 themes chosen and described enough, but with no way to assess face validity of this study.

We have described the process of analysis through which the themes emerged, and the process through which the themes were selected. 

  1. Para starting with line 244: This P has almost nothing to do with the UN card. Either remove or describe a bit why it is relevant.

We have described and interpreted why the narrative is relevant, connecting it to the theme.

  1. End of section 3.3: must mention that, on the other hand, being detained is HOW they have obtained the UN card.

Thank you. We have unpacked this in the section.

Discussion and/or other key points:

  1. Is there any validation of the story that it is more costly for healthcare (or true for any of the other 5 themes for that matter) for non-UN card holders? To be scientifically honest, it will help readers to understand the correlation between perception and accuracy. As another example, the validity of the belief that the UN card will get them to resettlement faster, than say applying for asylum immediately on arrival to Malaysia, is not explored. It sounds like, even with the UN card the process is long. I think they want it to be shorter, which has nothing to do with getting the UN card.

Yes, these perceptions are validated by empirical descriptions from other published sources. We have drawn in these sources and included them to validate the perceptions.

  1. Limitations: lack of showing the structure of all themes and a selection bias is highly likely. Also, these authors have published on their method and on this topic. How does this paper add to the good work the authors have already published?

Thank you. We have delineated how this work contributes and adds to the existing literature. We have also modified the limitations section.

Other specific comments:

  1. Lines 70-71 and 73-74 Redundant message, though clear

We have edited this.

  1. Starting line 398: The next three paragraphs are data and are redundant to the themes. If valuable, they should be in “Results.”

We have moved them to the results.

Please accept our acknowledgment for your extensive and encouraging comments.

Round 2

Reviewer 1 Report

I can definitely see that the paper has improved and gained in clarity.